# Transcriptome Analysis of Fibroblasts in Hypoxia-Induced Vascular Remodeling: Functional Roles of CD26/DPP4

**DOI:** 10.3390/ijms252312599

**Published:** 2024-11-23

**Authors:** Yuri Suzuki, Takeshi Kawasaki, Koichiro Tatsumi, Tadasu Okaya, Shun Sato, Ayako Shimada, Tomoko Misawa, Ryo Hatano, Chikao Morimoto, Yoshitoshi Kasuya, Yoshinori Hasegawa, Osamu Ohara, Takuji Suzuki

**Affiliations:** 1Department of Respirology, Graduate School of Medicine, Chiba University, Chiba 260-8670, Japan; 2Synergy Institute for Futuristic Mucosal Vaccine Research and Development, Chiba University, Chiba 260-8670, Japan; 3Department of Therapy Development and Innovation for Immune disorders and Cancers, Graduate School of Medicine, Juntendo University, Tokyo 113-8421, Japan; 4Department of Pharmacology, Faculty of Pharmacy, Juntendo University, Chiba 279-0013, Japan; 5Department of Applied Genomics, Kazusa DNA Research Institute, Chiba 292-0818, Japan

**Keywords:** hypoxia, vascular remodeling, fibroblast, CD26, dipeptidyl peptidase-4, DPP4, TGFβ

## Abstract

In hypoxic pulmonary hypertension (PH), pulmonary vascular remodeling is characterized by the emergence of activated adventitial fibroblasts, leading to medial smooth muscle hyperplasia. Previous studies have suggested that CD26/dipeptidyl peptidase-4 (DPP4) plays a crucial role in the pathobiological processes in lung diseases. However, its role in pulmonary fibroblasts in hypoxic PH remains unknown. Therefore, we aimed to clarify the mechanistic role of CD26/DPP4 in lung fibroblasts in hypoxic PH. *Dpp4* knockout (*Dpp4* KO) and wild-type (WT) mice were exposed to hypoxia for 4 weeks. The degree of PH severity and medial wall thickness was augmented in *Dpp4* KO mice compared with that in WT mice, suggesting that CD26/DPP4 plays a suppressive role in the development of hypoxic PH. Transcriptome analysis of human lung fibroblasts cultured under hypoxic conditions revealed that *TGFB2*, *TGFB3*, and *TGFA* were all upregulated as differentially expressed genes after *DPP4* knockdown with small interfering RNA treatment. These results suggest that CD26/DPP4 plays a suppressive role in TGFβ signal-regulated fibroblast activation under hypoxic conditions. Therefore, CD26/DPP4 may be a potential therapeutic target in patients with PH associated with chronic hypoxia.

## 1. Introduction

Acute alveolar hypoxia causes hypoxic pulmonary vasoconstriction, a homeostatic vasomotor response in resistant pulmonary arteries [1]. Chronic hypoxic stimulation leads to hypoxic pulmonary vascular remodeling in high-altitude residents, resulting in the development of hypoxic pulmonary hypertension (PH) [2]. Hypoxic PH is characterized by pathological and pathobiological changes in the blood vessel walls. Vascular medial smooth muscle hyperplasia develops in response to exposure to chronic hypoxia and is the main link between hypoxic PH and increased pulmonary vascular pressure [3,4], which is configured by apoptosis and an imbalance in proliferation in pulmonary artery smooth muscle cells (PASMCs) [5]. Hypoxia-inducible factor (HIF) plays a role in PASMCs’ proliferation and in the induction of various growth factors that are induced by hypoxia [6,7].

Pulmonary vascular remodeling in pulmonary arterial hypertension (PAH) is primarily characterized by intimal hyperplasia, medial hypertrophy, adventitial fibrosis, and inflammatory cell infiltration [8]. Chronic hypoxic exposure has been demonstrated to elicit structural changes in both the large proximal and distal muscular pulmonary arteries. The response to hypoxia is differentially regulated in specific cell types within vessel walls. Smooth muscle layers consist of multiple cell subpopulations in pulmonary blood vessels, and, in response to hypoxia, proliferative or matrix-producing responses occur in the larger ones [4]. In more peripheral vessels that are characterized by marked medial thickening in response to chronic hypoxia, a population of growth-resistant PASMCs has been observed [9], suggesting that factors other than resistant PASMCs could explain this marked medial thickening [4]. PH-associated pulmonary vascular remodeling is characterized by the emergence of an activated phenotype of pulmonary adventitial fibroblasts in hypoxic PH [10]. The earliest changes occur in the vascular adventitia [11,12]. Furthermore, it has been demonstrated that hypoxic stimulation affects the production of pro-inflammatory mediators and growth factors in fibroblasts and is involved in the recruitment of macrophages and the proliferation of PASMCs, while these responses are dependent on HIF [13,14]. In the systemic circulation, changes in the adventitia have been shown to precede other changes in several models of vascular injury [15,16], and adventitial fibroblasts may also play an important role in pulmonary circulation [17,18]; however, the response of fibroblasts to the development of PH remains unclear.

CD26/dipeptidyl peptidase-4 (DPP4) is a Type II membrane protein expressed on the surface of multiple cells [19]. The peptidase activity of CD26/DPP4 affects multiple proteins, including incretin hormones and has led to the development of CD26/DPP4 inhibitors as therapeutic agents against diabetes. CD26/DPP4 also participates in T cell activation and promotion of inflammation [20,21]. In human lungs, CD26/DPP4 is expressed in Type I and II alveolar epithelial cells, alveolar macrophages, and the vascular endothelium [22]. In addition, CD26/DPP4 plays a functional role in lung fibroblasts [23,24].

We have previously demonstrated that inhibition of CD26/DPP4 by the pharmaceutical agent sitagliptin ameliorates lipopolysaccharide (LPS)-induced lung injury in mice, with anti-inflammatory effects on the lung’s endothelial cells, and that CD26/DPP4 mediates inflammatory responses in the pulmonary endothelium [25,26,27]. In addition, using *Dpp4* knockout (*Dpp4* KO) mice, we have revealed the functional roles of CD26/DPP4 in bleomycin-induced PH associated with interstitial lung disease, bleomycin-induced pulmonary fibrosis, and LPS-induced lung injury [23,24,28]. In our bleomycin-induced pulmonary fibrosis study, the numbers of fibroblasts and myofibroblasts in *Dpp4* KO mouse lungs after bleomycin treatment were lower than those in wild-type (WT) mice, and the suppression of CD26/DPP4 expression by the small interfering RNA (siRNA) treatment had inhibitory effects on lung fibroblasts’ activation in vitro [24]. These findings suggest a plausible role of CD26/DPP4 in lung fibroblast-related pathobiology. However, the functional role of CD26/DPP4 in lung fibroblasts in hypoxic PH remains unknown.

In this study, we aimed to elucidate the functional role of CD26/DPP4 in chronic hypoxic PH by focusing on lung fibroblasts. We established mouse models of chronic hypoxic PH using WT and *Dpp4* KO mice and performed a transcriptome analysis of human lung fibroblasts (HLFs) cultured under hypoxic conditions with *DPP4* knockdown (*DPP4* KD) by siRNA treatment.

## 2. Results

### 2.1. Hypoxia-Induced Pulmonary Hypertension Was Augmented in Dpp4 KO Mice

To investigate the functional roles of CD26/DPP4 in the pathogenesis of PH during chronic hypoxia, the degree of PH was compared between WT and *Dpp4* KO mice housed under normoxic or chronic hypoxic conditions. Both right ventricular (RV) systolic pressure (RVSP) and Fulton’s index were higher in chronic hypoxic WT mice than in normoxic WT mice (*p* < 0.001). Although RVSP and Fulton’s index were not significantly different between WT and *Dpp4* KO mice housed under normoxic conditions, RVSP was significantly higher in *Dpp4* KO mice than in WT mice housed under chronic hypoxic conditions (*p* < 0.001) (Figure 1a). No significant difference was observed in Fulton’s index under hypoxic conditions, but it tended to be higher in *Dpp4* KO mice than in WT mice (*p* = 0.13) (Figure 1b).

### 2.2. Hypoxia-Induced Medial Wall Thickness Was Augmented in Dpp4 KO Mice

The medial wall thickness of the small pulmonary vessels exposed to either normoxia or hypoxia was compared between WT and *Dpp4* KO mice. While no differences were observed between the WT and *Dpp4* KO mice housed under normoxic conditions (Figure 2a,b), the medial wall thickness of the small pulmonary vessels tended to increase in chronically hypoxic WT mice compared with normoxic WT mice (*p* = 0.086). Additionally, this effect was significantly greater in *Dpp4* KO mice than in WT mice under chronic hypoxic conditions (*p* < 0.0001) (Figure 2a,b). These data suggest that chronic hypoxia caused an increase in the medial wall thickness, and this response was further enhanced in *Dpp4* KO mice.

### 2.3. The Number of Lung Endothelial Cells Was Larger During Chronic Hypoxia, and Dpp4 KO Did Not Affect This Response

We assessed the effects of chronic hypoxic exposure on the lung’s constituent cells in WT and *Dpp4* KO mice using flow cytometry (FCM). Although the number of lung endothelial cells was larger in chronically hypoxic mice than in normoxic mice, no significant differences were observed between chronically hypoxic WT and *Dpp4* KO mice (Figure 3a). Additionally, no significant differences were observed in other constituent cells of the lung, such as epithelial, mesenchymal, and hematopoietic cells, regardless of *Dpp4* KO under hypoxic conditions (Figure 3b–d).

### 2.4. Deletion of Dpp4 Does Not Significantly Affect Inflammatory Responses During Acute or Subacute Hypoxia

To determine whether inflammatory responses are associated with the development of hypoxic PH, bronchoalveolar lavage fluid (BALF) neutrophils and BALF protein levels were examined in the control (baseline), acute (Days 1 and 7), and subacute (Day 14) phases in WT/hypoxic and *Dpp4* KO/hypoxic mice. We hypothesized that neutrophilic inflammatory responses would be involved in the subacute phase during the development of hypoxic PH. However, no significant differences in the number of neutrophils and protein levels in BALF were observed between WT/hypoxic and *Dpp4* KO/hypoxic mice on Days 1, 7, and 14 (Figure 4a,b).

### 2.5. Transcriptome Analysis of Cultured Human Lung Fibroblasts (HLFs)

To explore the potential effects of hypoxia on the transcriptome signature of cultured HLFs, we compared HLFs cultured under normoxic conditions (control/normoxia) and those cultured under hypoxic conditions (control/hypoxia) treated with negative control siRNA. Next, to examine the effects of *DPP4*-siRNA treatment on the transcriptome signature of cultured HLFs, we compared normoxia-cultured HLFs treated with negative control siRNA (control/normoxia) with those treated with *DPP4*-siRNA (*DPP4* KD/normoxia). Additionally, to investigate the effects of *DPP4*-siRNA under hypoxic conditions on the transcriptome signature of cultured HLFs, we compared hypoxia-cultured HLFs treated with negative control siRNA (control/hypoxia) with those treated with *DPP4*-siRNA (*DPP4* KD/hypoxia). Treatment with *DPP4*-siRNA significantly downregulated the mRNA expression levels of *DPP4* in HLFs (Appendix A).

#### 2.5.1. Differential Gene Expression and Pathway Analysis Between HLFs Cultured Under Normoxic Conditions and Those Cultured Under Hypoxic Conditions

RNA sequencing libraries were prepared from mRNA isolated from normoxia- and hypoxia-cultured HLFs. The RNA integrity values of all samples were >9, and 26,467 genes were identified from the mRNA. Additional quality control methods were used to compare the normoxia- and hypoxia-cultured HLFs, and 13,118 genes were retained for further analysis.

Principal component analysis (PCA) revealed that the two groups could be distinguished (Figure 5a). We compared differentially expressed genes (DEGs) between normoxia- and hypoxia-cultured HLFs. Figure 5b shows a volcano plot of the distribution of the log2 fold changes and *p*-values for 13,118 genes expressed in these samples. Of these, 257 genes were identified as DEGs (*p* < 0.01, fold change >3 or <0.33). Figure 5c shows a heat map of the 257 DEGs, with 191 upregulated and 66 downregulated genes in hypoxia-cultured HLFs compared with in normoxia-cultured HLFs. Details of the DEGs are provided in Appendix A.

Enrichment analysis using the Enrichr online tool revealed that some gene ontologies (GOs) and pathways were significantly enriched in the DEGs between normoxic and hypoxic HLFs. GO terms for the upregulated genes are listed in Table 1, 1a. Kyoto Encyclopedia of Genes and Genomes (KEGG) pathway terms for upregulated genes are listed in Table 1b.

Several terms related to cellular responses to hypoxia and the p38 mitogen-activated protein kinase (p38 MAPK) cascade were enriched in the GO and KEGG pathway analyses.

#### 2.5.2. Differential Gene Expression and Pathway Analysis Between Control HLFs and Those with DPP4 KD Cultured Under Normoxic Conditions

RNA sequencing libraries were prepared from mRNA isolated from the control and *DPP4* KD HLFs under normoxic conditions. The RNA integrity values of all samples were >9, and 26,467 genes were identified from the mRNA. Additional quality control methods were used to compare control and *DPP4* KD HLFs under normoxic conditions, and 13,032 genes were retained for further analysis.

PCA revealed that the two groups could be distinguished (Figure 5d). We compared the DEGs in the control and *DPP4* KD HLFs under normoxic conditions. Figure 5e shows a volcano plot of the distribution of the log2 fold changes and *p*-values for the 13,032 genes expressed in these samples. Of these, 557 genes were identified as DEGs (*p* < 0.01, fold change >3 or <0.33). Figure 5f shows a heat map of the 557 DEGs, with 279 upregulated and 278 downregulated genes in *DPP4* KD HLFs under normoxic conditions compared with the control HLFs. Details of the DEGs are provided in Appendix A.

Enrichment analysis using the Enrichr online tool revealed that some GOs and pathways were significantly enriched in the DEGs between the control and *DPP4* KD HLFs under normoxic conditions. GO terms for the upregulated genes are listed in Table 2a. KEGG pathway terms for the upregulated genes are listed in Table 2b.

Several terms related to cellular responses to hypoxia, the extracellular matrix (ECM), and intracellular signaling were enriched using GO and KEGG pathway analyses.

#### 2.5.3. Differential Gene Expression and Pathway Analysis Between Control and DPP4 KD HLFs Under Hypoxic Conditions

RNA sequencing libraries were prepared from mRNA isolated from the control and *DPP4* KD HLFs under hypoxic conditions. The RNA integrity values of all samples were >9, and 26,467 genes were identified from the mRNA. Additional quality control methods were used to compare the control and *DPP4* KD HLFs under hypoxic conditions, and 13,307 genes were retained for further analysis.

PCA revealed that the two groups could be distinguished (Figure 5g). We compared the DEGs in the control and *DPP4* KD HLFs under hypoxic conditions. Figure 5h shows a volcano plot of the distribution of the log2 fold changes and *p*-values for the 13,307 genes expressed in these samples. Of these, 658 genes were identified as DEGs (*p* < 0.01, fold change >3 or <0.33). Figure 5i shows a heat map of the 658 DEGs, with 337 upregulated and 321 downregulated genes in hypoxic *DPP4* KD HLFs compared with the control HLFs. Details of the DEGs are provided in Appendix A.

Enrichment analysis using the Enrichr online tool revealed that some GOs and pathways were significantly enriched in the DEGs between the control and *DPP4* KD HLFs under hypoxic conditions. GO terms for upregulated genes are listed in Table 3a. KEGG pathway terms for the upregulated genes are listed in Table 3b.

Several terms related to cellular responses to hypoxia, transforming growth factor (TGF) β, ECM, and intracellular signaling were enriched using GO and KEGG pathway analyses.

## 3. Discussion

In this present study, to investigate whether CD26/DPP4 is involved in the pathogenesis of PH during chronic hypoxia, hemodynamic and pathological analyses were performed using *Dpp4* KO mice. This present study demonstrated that chronic hypoxic exposure promotes the development of PH, and the repression of *Dpp4* expression further increases the severity of PH under chronic hypoxic conditions. These findings indicated that CD26/DPP4 exerts protective effects against the development of PH during hypoxic exposure. Enrichment analysis using the transcriptome of HLFs cultured under hypoxic conditions revealed that terms with genes upregulated by *DPP4* KD included “regulation of TGF-beta production”, “regulation of TGF-beta2 production”, and “extracellular matrix organization” among the relevant GO terms and “TGF-beta signaling pathway” among the relevant KEGG pathway terms. *TGFB2*, *TGFB3*, and *TGFA* were all DEGs that were upregulated by *DPP4* KD in HLFs cultured under hypoxic conditions, suggesting that CD26/DPP4 exerts suppressive effects on fibroblast activation via the TGFβ signaling pathway in chronic hypoxic conditions.

This study showed that the degree of PH and medial wall thickness after exposure to chronic hypoxia was greater in *Dpp4* KO mice than in WT mice, indicating that an inherent defect of *Dpp4* leads to the deterioration of hypoxic PH, and the presence of *Dpp4* suppresses medial wall thickness in PH (Figure 1 and Figure 2). While we also observed an increase in endothelial cell numbers in the hypoxic PH model, as reported previously [29], this response was not affected by *DPP4* KD (Figure 3a). CD26/DPP4 is expressed in various constituent cells of the lung, such as Type I and II alveolar epithelial cells, alveolar macrophages, vascular endothelial cells, and lung fibroblasts [22,23,24]. CD26/DPP4 inhibitors have been used clinically as therapeutic agents against diabetes, which can be associated with pulmonary hypertension [30]. It has been reported that the administration of sitagliptin, a DPP4 inhibitor, alleviated chronic hypoxia-induced PH in rats and cultured human PASMCs, and DPP4 inhibitors suppressed platelet-derived growth factor (PDGF)–BB-induced proliferation and migration [31]. In addition, that study showed that the inhibition of DPP4 by sitagliptin alleviated hemodynamic and histological changes in monocrotaline (Mct)-induced models of experimental PH in rats, and a DPP4 inhibitor inhibited Mct-induced fibrosis in the adventitia of the pulmonary artery [31]. Remodeling of the pulmonary arterial adventitia in PH is characterized by extensive fibrosis and inflammatory cell infiltration. Extensive pulmonary arterial adventitial fibrosis in Mct-treated rats was reduced by administration of sitagliptin. Sitagliptin inhibits the perivascular accumulation of inflammatory cells, including mast cells and macrophages [31]. These observations suggest that the beneficial effect of DPP4 inhibitors on pulmonary vessel remodeling in PH could, in part, be achieved by the prevention of adventitial fibrosis and inflammatory cell infiltration. Taken together with the findings of this present study, the effects of genetic and pharmacological DPP4 inhibition could differ in the pathogenesis of vasculopathy in PH, depending on the causes and/or species, and further studies are required.

In this study, we explored the functional role of CD26/DPP4 in lung fibroblasts using a hypoxic PH model. Transcriptome data showed that *TGFB2*, *TGFB3*, and *TGFA* were all upregulated by *DPP4* KD in HLFs cultured under hypoxic conditions, suggesting that CD26/DPP4 plays a suppressive role in the TGFβ signal-induced fibroblast activation under chronic hypoxic conditions (Table 3). On the contrary, our previous study indicated that CD26/DPP4 suppression is implicated in the attenuation of pulmonary fibrosis and decreases in the numbers of fibroblasts and myofibroblasts through a TGFβ-dependent manner in BLM-induced lung fibrosis in mice [24]. CD26/DPP4 expression has been shown in the fibroblast lineage involved in the chronic fibrotic status of many organs, such as skin fibrosis in systemic sclerosis, liver cirrhosis, kidney fibrosis, and lung fibrosis [32,33]. Upregulation of CD26/DPP4 promotes the activation of TGFβ pathways in fibroblasts and the TGFβ signaling pathway, leading to the activation of fibroblasts [32]. Taken together, the functional roles of CD26/DPP4 may vary depending on the cellular situation, resulting in varying degrees of PH.

In this present study, the integrin β8 subunit (*ITGB8*) gene was upregulated by *DPP4* KD under hypoxic conditions and was included in the GO terms, such as “regulation of TGF-beta production” (Table 3). The inhibition of the DPP4–integrin β interaction and TGFβ signal transduction has been reported to be related to antifibrotic effects exerted by DPP4 inhibitors. In the kidneys, the interaction of CD26/DPP4 and integrin β1 facilitates renal fibrosis by enhancing TGFβ receptor-mediated endothelial–mesenchymal transition [34]. Interleukin-1β (IL-1β), a crucial pro-inflammatory cytokine implicated in lung fibrosis, increases *ITGB8* expression in the lung fibroblasts, which increases αvβ8-mediated TGFβ activation in fibrosis and pathologic inflammation. IL-1β, through p38-MAPK and nuclear factor-κB-dependent pathways, strengthens the association of the activator protein-1 complex members (c-Jun and ATF-2) with the *ITGB8* core promoter region [35]. In this present study, the “MAPK signaling pathway” was included in the KEGG pathway terms, with several genes upregulated by *DPP4* KD in HLFs under hypoxic conditions (Table 3). Taken together, these findings suggest that the upregulation of integrin β8 subunit (*ITGB8*) by *DPP4* KD in fibroblasts under hypoxic conditions is responsible for the upregulation of the TGFβ signaling pathway.

The elastin (*ELN*) gene was upregulated by *DPP4* KD under hypoxic conditions and was included in the GO terms as “extracellular matrix organization” (Table 3). *ELN* encodes a protein of elastic fibers, which comprise part of the extracellular matrix and give elasticity to the lungs and blood vessels. Elastin is an important component of the lung interstitium that facilitates the recoil of the pulmonary vasculature. Upregulation of elastin synthesis is associated with the activation of TGFβ, and disruption of elastin results in subendothelial proliferation of smooth muscle cells, thus contributing to vascular remodeling [36]. The degradation products of elastin, known as elastin-derived peptides, bind to the elastin receptor complex and stimulate the migration and proliferation of monocytes and fibroblasts. The reduction in CD26/DPP4 levels via siRNA treatment in HLFs cultured under hypoxic conditions could be associated with a decrease in the extracellular matrix’s organization. Thus, CD26/DPP4 might play a suppressive role in activated lung fibroblasts in hypoxic PH.

The “MAPK signaling pathway” was a relevant enriched KEGG term, including *TGFB2* and *TGFB3*, which were upregulated by *DPP4* KD HLFs cultured under hypoxic conditions (Table 3). It has been demonstrated that hypoxia induces adventitial hypertrophy of the pulmonary arteries with extracellular matrix accumulation in chronic hypoxic rats, and hypoxia upregulates the expression of α-smooth muscle actin, collagen Type I-A, and fibronectin in cultured fibroblasts potentially through an increase in 15-lipoxygenase/15-hydroxyeicosatetraenoic acid (15-LO/15-HETE). The 15-LO/15-HETE-induced fibrosis in the adventitia and alterations in the fibroblast phenotype occur through TGFβ1/Smad2/3 pathway signaling. Additionally, p38 MAPK potentially mediates 15-LO-induced TGFβ1 and Smad2/3 activation after hypoxia [37]. Therefore, adventitial fibrosis was suggested to be an essential event in chronic hypoxic PH, via the 15-LO/15-HETE-p38 MAPK-TGFβ1/Smad2/3 intracellular signaling systems. The reduction in CD26/DPP4 levels via siRNA treatment in HLFs cultured under hypoxic conditions was associated with the activation of the MAPK signaling pathway; thus, CD26/DPP4 in lung fibroblasts may play a role in the inhibition of adventitial fibrosis in hypoxic PH.

CD26/DPP4 is widely expressed in various cell types in lung tissues, such as Type I and II alveolar cells, alveolar macrophages, and vascular endothelial cells, and has recently been suggested as a therapeutic target in lung diseases [19,23,24,25,27,28]. In this current study, enrichment analysis of the HLF transcriptome under normoxic conditions revealed many DEGs following *DPP4* KD (Table 2 and Appendix A). One of the relevant GO terms associated with genes upregulated by *DPP4* KD, including DEGs of *SFRP1* (Secreted Frizzled related protein 1 gene), *HMOX1* (heme oxygenase 1 gene), *HIF1A* (Hypoxia inducible factor-1α gene), and *AQP1* (Aquaporin 1 gene) was “cellular response to hypoxia”. A recent study demonstrated that, in HLFs and murine lung regeneration tissues, Sfrp1-positive cells appeared early after lung injury in adventitial, peribronchiolar, and alveolar locations and preceded the emergence of myofibroblasts, suggesting *SFRP1* as a modulator of TGFβ1-driven fibroblast phenotypes in fibrogenesis [38]. *HMOX1* encodes for the enzyme heme oxygenase-1 (HO-1), an oxygen-dependent enzyme which regulates vascular tone and cell proliferation in the pulmonary circulation. In a rat model of chronic hypoxia, HO-1 was induced during PASMCs’ proliferation [39]. It has been reported that HIF subtypes (HIF-1α, HIF-2α, and HIF-3α) and the dependent target genes, carrying the hypoxia-responsive element as a regulatory component, were strongly activated in both adventitial fibroblasts and smooth muscle cells under hypoxic conditions. Therefore, hypoxia-driven gene regulation in pulmonary artery fibroblasts results in a mitogenic stimulus in the adjacent PASMCs [13]. *AQP1* is related to the profibrotic TGFβ action and plays a role in the profibrotic transformation to a mesenchymal phenotype (myofibroblasts) induced by TGFβ in fibroblasts in vitro [40]. The relevant GO term of “cellular response to hypoxia” associated with genes upregulated by *DPP4* KD was commonly included under both normoxic and hypoxic conditions. Taken together, the suppression of CD26/DPP4 in HLFs potentially enhances the upregulation of hypoxic responses through *SFRP1*, *HMOX1*, *HIF1A*, and *AQP1* regardless of the oxygen concentration, and is related to fibroblast activation in hypoxic PH.

In this study, the “cGMP-PKG signaling pathway” was included as a term of upregulated genes following *DPP4* KD in both normoxia and hypoxia (Table 2 and Table 3). Among the phosphodiesterase (PDE) enzymes, which degrade cGMP, *PDE5A* was detected as a DEG downregulated by *DPP4* KD in HLFs under normoxic and hypoxic conditions (Appendix A). PDE is suggested to contribute to pulmonary fibrosis, and there is a report stating that the PDE4B inhibitor suppress pulmonary fibrosis in mice and HLFs from patients with idiopathic pulmonary fibrosis (IPF) by inhibiting the TGFβ signaling pathway and suppressing cell proliferation via IL-1β [41]. Furthermore, clinical studies have indicated that it is effective in treating patients with IPF [42]. An in vitro fibroblast study also reported that the PDE5 inhibitor vardenafil may improve pulmonary fibrosis via TGFβ1 [43]. However, the role of PDE5 in fibroblasts in hypoxic PH is unknown, and further research is required.

Reduced PPARγ activity has been proven to contribute to PAH via the TGFβ signaling pathway [44,45], and we also focused on its expression in HLFs. However, it was not detected as a DEG regardless of *DPP4* KD and/or hypoxic exposure (Appendix A).

Our study had several limitations. First, we focused on lung fibroblasts in the pathogenesis of hypoxic PH, whereas other cells that compose the pulmonary vascular wall, such as endothelial, smooth muscle, and inflammatory cells, were not evaluated. It has been suggested that the intracellular responses of these cells and their interactions are involved in the development of hypoxic PH. Therefore, global KO mice limit the ability to characterize the cell-specific roles of CD26/DPP4 in vivo, and our results from mouse model experiments may not directly reflect those from in vitro fibroblast studies. Second, only female mice were used. Previous studies have reported sex differences in some types of PH and a relationship between estrogen receptors and HIF expression [46]. Thus, the potential sex differences remain unknown. Third, the expression and function of CD26/DPP4 in patients with hypoxic PH has not been evaluated in clinical practice. Lastly, to explore the translational potential of DPP4 inhibitors, additional studies in hypoxic PH mouse models are warranted to assess their therapeutic efficacy. Further research is required to better understand the functional roles of CD26/DPP4 in the pathogenesis of hypoxia-induced pulmonary vascular remodeling.

## 4. Materials and Methods

### 4.1. Animal Model of Hypoxic Pulmonary Hypertension

All animal protocols were approved by the Review Board for Animal Experiments of Chiba University (Chiba, Japan). Seven-to-nine-week-old female C57BL/6J mice (Japan SLC Inc., Shizuoka, Japan) were used as wild-type (WT) mice. *Dpp4* KO mice with a C57BL/6 background were kindly provided by the laboratory of Dr. Chikao Morimoto at Juntendo University (Tokyo, Japan) and were bred and housed in our facility. To establish the hypoxia-induced PH model, the WT and *Dpp4* KO mice were housed under normoxic (room air, 21% O_2_) or hypoxic (10% O_2_) conditions in the same room for 4 weeks (range: 28–29 days). To evaluate the effects of acute and subacute hypoxic exposure, we established mouse models exposed to hypoxia for 1, 7, or 14 days.

### 4.2. Hemodynamic Analysis

RVSP, which reflects the pulmonary arterial systolic pressure, was evaluated using right heart catheterization. After exposure to hypoxia for 4 weeks, the mice were mildly anesthetized with inhaled isoflurane (3% for induction, 1% for maintenance) while maintaining spontaneous breathing and their body temperatures at 37 °C. A 1.4 F microtip pressure transducer catheter (SPR-671; Millar OEM Solutions, Houston, TX, USA) was inserted into the right jugular vein through a small incision and passed through the RV. A Power-Lab data acquisition system (AD Instruments, Dunedin, New Zealand) was used to record the RVSP.

### 4.3. Fulton’s Index

The hearts were removed from the mice, and the RV was carefully dissected from the left ventricle (LV) and septum. To evaluate RV hypertrophy, Fulton’s index was calculated as the ratio of the weight of the RV to that of the LV plus the septum.

### 4.4. Histological Analysis

Mouse lungs were inflation-fixed with 4% paraformaldehyde, embedded in paraffin, sectioned, mounted on slides, and stained with EVG. To assess pulmonary vascular remodeling, the medial wall thickness of the small pulmonary vessels was calculated as the average thickness of four medial walls divided by the average diameter of two perpendicular external elastic laminae [47].

### 4.5. Collection of BALF

After acute or prolonged hypoxic exposure, BALF was collected by instilling 1 mL of phosphate-buffered saline through a tracheal cannula into the lungs, followed by slow recovery of the fluid. Cells were counted using an automated cell counter (TC20; Bio-Rad, Hercules, CA, USA) and collected by centrifugation (500× *g*, 20 min, 4 °C). The BALF protein concentration was measured using a bicinchoninic acid assay kit (Thermo Fisher Scientific, Waltham, MA, USA).

### 4.6. Flow Cytometry Analysis of Mouse Cells in BALF and Lungs

Mouse cells in BALF or lung tissues were pretreated with anti-mouse CD16/32 (BioLegend, San Diego, CA, USA) for 10 min to block Fc receptors and then incubated with specific antibodies in the dark at 4 °C for 15 min. The following antibodies were used for cell-surface staining: anti-mouse CD31-PE/Cy7 (BioLegend), anti-mouse CD45-Alexa Fluor 700 (BioLegened), anti-mouse CD326-PerCP/Cyanine5.5 (BioLegend), anti-mouse CD26-FITC (BioLegend), and anti-mouse Gr-1-APC (BioLegend). The cell fluorescence was measured using a BD FACS Canto II system (BD Biosciences, Franklin Lakes, NJ, USA), and data were analyzed using FlowJo software v10.7.1 (BD Biosciences). CD26/DPP4 expression levels were evaluated in the cellular components of the mouse lungs using FCM. CD26/DPP4 levels were confirmed to be substantially low or nearly zero in *Dpp4* KO mice (Appendix A).

### 4.7. Cell Culture and Treatments of Small Interfering RNA

To explore the potential effects of CD26/DPP4 on HLFs under hypoxic conditions, we performed RNA sequencing of HLFs treated with negative control siRNA (control) or *DPP4*-siRNA (*DPP4* KD) cultured under normoxic or hypoxic conditions.

HLFs were obtained from Lonza (Basel, Switzerland) and cultured in Dulbecco’s Modified Eagle’s Medium with high glucose supplemented with 15% fetal bovine serum. For siRNA transfection, *DPP4*-siRNA (Cat# 4392421, siRNA ID:s4255) and negative control siRNA (Cat# 4390843, Silencer™ Select Negative Control No.1 siRNA) were purchased from Thermo Fisher Scientific. Using the Lipofectamin™ RNAiMAX Transfection Reagent (Thermo Fisher Scientific), HLFs were transfected with siRNA for 72 h according to the manufacturer’s protocol. After siRNA treatment, the cells were exposed to normoxia or hypoxia. Cells in the normoxic groups were maintained at 37 °C in 21% O_2_ and 5% CO_2_. Cells in the hypoxia groups were separately cultured in 1% O_2_ and 5% CO_2_ for 48 h, using AnaeroPack-Kenki 5% (Mitsubishi Gas Chemical, Tokyo, Japan) and an oxygen indicator. Cells were used at Passages 6–8 in all experiments.

In this study, we used the same HLFs and *DPP4*-siRNA as used in a previous study [24]. Treatment with *DPP4*-siRNA significantly downregulated the mRNA expression levels of *DPP4* in HLFs (Appendix A).

### 4.8. Transcriptome Analysis

Total RNA was isolated from HLFs and stored in Isogen (Nippon Gene, Tokyo, Japan). Subsequently, 1 mL of this solution was vigorously vortexed and then centrifuged after adding 200 μL of chloroform. The upper aqueous layer was collected, and 10 µg of glycogen (Roche, Basel, Switzerland) was then added. RNA was precipitated by adding 500 µL of isopropyl alcohol. The solution was then vortexed vigorously and centrifuged. The RNA pellets were washed with 75% ethanol and then dissolved in 10 µL of RNase-free water. The concentration and quality of RNA were verified using an Agilent 2100 Bioanalyzer (Agilent Technologies, Santa Clara, CA, USA). Purified total RNA (200 ng) with an RNA integrity value of > 9 was used for RNA library preparation according to the instructions of the QuantSeq 3′mRNA-Seq Library Prep Kit FWD for Illumina (Lexogen, Vienna, Austria). Libraries were amplified via 13 cycles of polymerase chain reaction. RNA libraries were sequenced using the Illumina (San Diego, CA, USA) NextSeq 500 system (75 cycles) with NextSeq System Suite v4.2.0. The FASTQ files were prepared with reads using bcl2fastq ver2.20 (Illumina). The quality of the FASTQ sequence data was assessed using FastQC v0.11.9 (Illumina). After removing adapter sequences from the raw reads, the trimmed reads were aligned to the GRCh38 human reference genome using STAR v2.7.6a. Reads per million (RPM) values were calculated using Samtools v1.15 and htseq count v1.99.2. The expression levels of the genes identified in the transcriptome were normalized and compared.

### 4.9. mRNA Sequencing Data Analysis

The RPM data were log2-transformed and filtered to ensure that at least one group contained at least 70% valid values for each gene. The remaining missing values were imputed by using random numbers drawn from a normal distribution (width = 0.3, downshift = 2.8). The unpaired Student’s t-test was used to compare the two groups. Statistical significance was defined as a two-sided *p*-value of < 0.05. The false discovery rate was also calculated as a q-value and used to interpret the *p*-values.

RPM data were analyzed using PCA, and heat maps with hierarchical clustering were drawn using the Qlucore Omics Exploration software ver. 3.9.9 (Qlucore AB, Lund, Sweden). The fold changes between each group (>3.0 upregulated or <0.33 downregulated) with *p*-values of <0.01 were set to detect DEGs. Significantly over-represented functional categories were identified using the Enrichr online tool (http://amp.pharm.mssm.edu/Enrichr/), accessed on 13 June 2024. The gene set databases used in this study were “Gene Ontology (GO)_Biological_Process_2023” (terms: 5047; gene coverage: 14,698) and “Kyoto Encyclopedia of Genes and Genomes (KEGG)_2021_Human” (terms: 320; gene coverage: 8078). GO terms and KEGG pathways were considered statistically significant at *p* < 0.05.

Genes with significantly upregulated expression between control/normoxia and control/hypoxia, between control/normoxia and *DPP4* KD/normoxia, or between control/hypoxia and *DPP4* KD/hypoxia were annotated.

### 4.10. Statistical Analysis

The results were expressed as the means ± standard deviation (SD). When the analysis group was assumed to be normally distributed according to the Shapiro–Wilk test, two-way analysis of variance was used for multiple group comparisons, followed by Tukey’s post hoc test. Unpaired two-tailed *t*-tests were used to compare the two groups. If the data did not have a normal distribution, the Kruskal–Wallis test was performed. Statistical significance was set at *p* < 0.05. Statistical analyses were performed using GraphPad Prism version 9.5.1 (GraphPad Inc., La Jolla, CA, USA).

## 5. Conclusions

This study demonstrated that the loss of CD26/DPP4 augments hypoxic PH in mice by exacerbating vascular remodeling, and CD26/DPP4 may play a suppressive role in fibroblast activation via the TGFβ signaling pathway in chronic hypoxic conditions. Therefore, CD26/DPP4 may be a potential therapeutic target in patients with PH associated with chronic hypoxia.

## Figures and Tables

**Figure 1 ijms-25-12599-f001:**
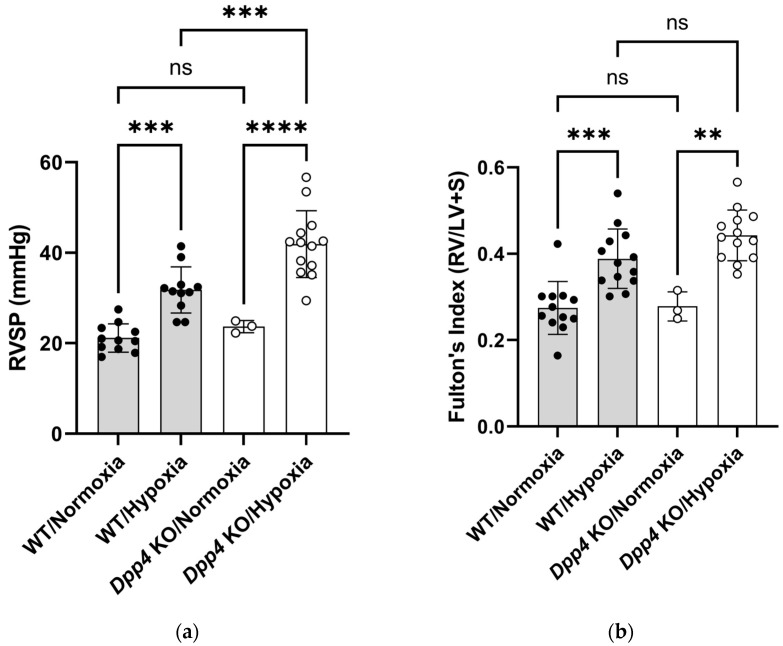
Pulmonary hemodynamic evaluation of hypoxia-induced pulmonary hypertension in wild-type (WT) and *Dpp4* knockout (*Dpp4* KO) mice. (**a**) After 4 weeks, right ventricular systolic pressure (RVSP) was significantly higher in WT/hypoxia mice than in WT/normoxia mice, and in *Dpp4* KO/hypoxia mice than in WT/hypoxia mice. (**b**) Fulton’s index (the weight ratio of the right ventricle to the left ventricle plus the ventricular septum) was significantly higher in WT/hypoxia mice than in WT/normoxia mice, and *Dpp4* KO/hypoxia mice tended to have a higher index than WT/hypoxia mice (*p* = 0.13). ns; not significant, ** *p* < 0.01, *** *p* < 0.001, **** *p* < 0.0001.

**Figure 2 ijms-25-12599-f002:**
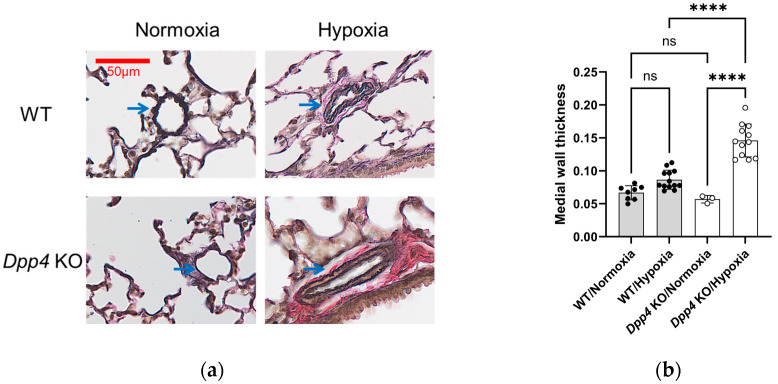
Evaluation of pulmonary small vessel remodeling in hypoxia-induced pulmonary hypertension in WT and *Dpp4* KO mice. (**a**) Representative Elastica van Gieson (EVG)-stained small pulmonary vessels in mice. Blue arrows indicate pulmonary small vessels. Scale bar, 50 μm. (**b**) The medial wall thickness of the small pulmonary vessels in mice was calculated as the average thickness of four medial walls divided by the average diameter of two perpendicular external elastic laminae. ns; not significant, **** *p* < 0.0001.

**Figure 3 ijms-25-12599-f003:**
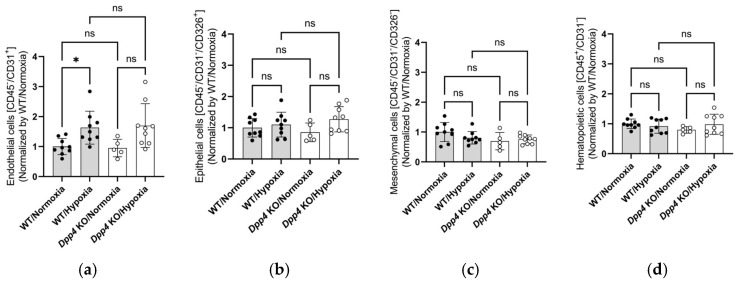
The number of constituent cells of the lung in *Dpp4* KO or WT mice 4 weeks after normoxic or hypoxic exposure evaluated by flow cytometry (*n* = 5–9). (**a**) The number of CD45^−^/CD31^+^ endothelial cells was larger during chronic hypoxia, although *Dpp4* KO did not affect this response. (**b**–**d**) No significant differences were observed in the numbers of CD45^−^/CD31^−^/CD326^+^ epithelial cells (**b**), CD45^−^/CD31^−^/CD326^−^ mesenchymal cells (**c**), and CD45^+^/CD31^−^ hematopoietic cells (**d**) regardless of *Dpp4* KO under hypoxic conditions. ns; not significant, * *p* < 0.05.

**Figure 4 ijms-25-12599-f004:**
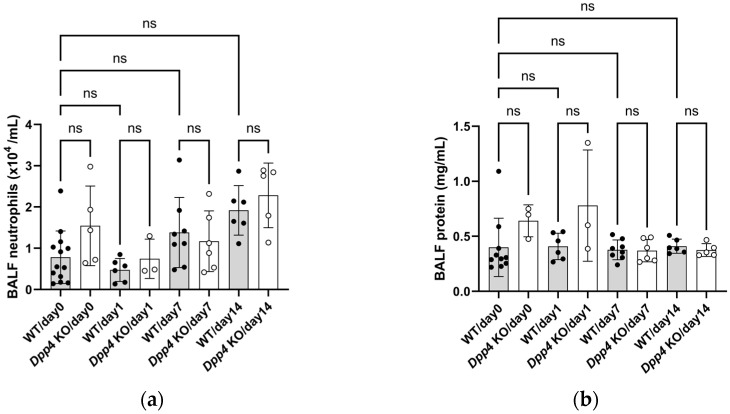
Inflammatory responses during acute and subacute hypoxic exposure. (**a**) The number of CD45^+^/Gr-1^+^ neutrophils in bronchoalveolar lavage fluid (BALF) from the mice was not different between WT/hypoxic and *Dpp4* KO/hypoxic mice. (**b**) BALF protein levels did not differ between WT/hypoxic and *Dpp4* KO/hypoxic mice. ns; not significant.

**Figure 5 ijms-25-12599-f005:**
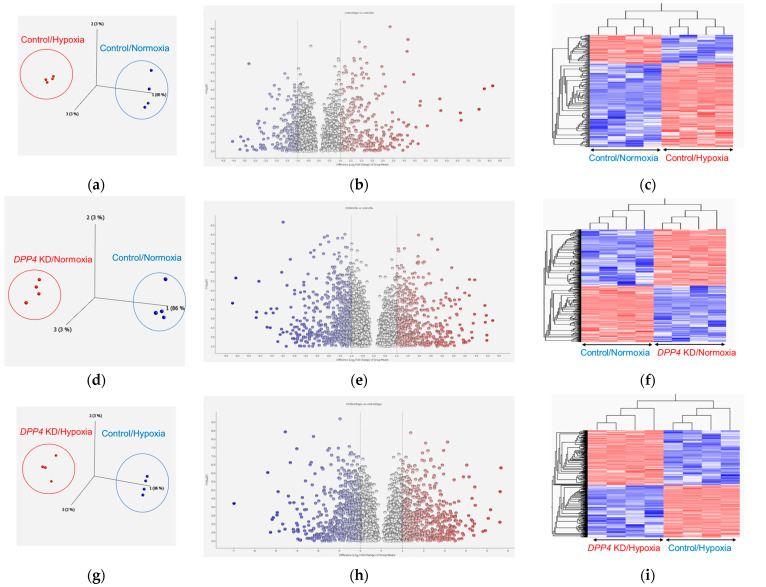
Transcriptome analysis of cultured human lung fibroblasts (HLFs) treated with hypoxia and *DPP4*-small interfering RNA (siRNA). Cultured HLFs were treated as follows (each group, n = 4): Control/normoxia (treated with negative control siRNA followed by exposure to normoxic conditions), control/hypoxia (treated with negative control siRNA followed by exposure to hypoxic conditions), *DPP4* KD/normoxia (HLFs treated with *DPP4*-siRNA followed by exposure to normoxic conditions),and *DPP4* KD/hypoxia (HLFs treated with *DPP4*-siRNA followed by exposure to hypoxic conditions). (**a**–**c**) Comparisons between control/normoxia and control/hypoxia: (**a**) principal component analysis (PCA) revealed that the two groups could be distinguished; (**b**) volcano plot of the distribution of the log2 fold changes and *p*-values, with blue dots representing downregulated differentially expressed genes (DEGs) and red dots representing upregulated DEGs; (**c**) heat map of the DEGs. (**d**–**f**) Comparisons between control/normoxia and *DPP4* KD/normoxia: (**d**) PCA revealed that the two groups could be distinguished; (**e**) volcano plot of the distribution of the log2 fold changes and *p*-values, with blue dots representing downregulated DEGs and red dots representing upregulated DEGs; (**f**) heat map of the DEGs. (**g**–**i**) Comparisons between control/hypoxia and *DPP4* KD/hypoxia: (**g**) PCA revealed that the two groups could be distinguished; (**h**) volcano plot of the distribution of the log2 fold changes and *p*-values, with blue dots representing downregulated DEGs and red dots representing upregulated DEGs; (**i**) heat map of the DEGs.

**Table 1 ijms-25-12599-t001:** Enrichment analysis of transcriptomic data (normoxia- vs. hypoxia-cultured HLFs).

(a) Excerpted Relevant GO Terms		
Terms with Upregulated Genes Following Exposure to Hypoxia	*p*-Value	Genes
Cellular response to hypoxia (GO:0071456)	<0.001	*SLC8A3*, *BNIP3L*, *EGLN3*, *PTGIS*, *HILPDA*, *RORA*, *AK4*, *NDNF*, *NDRG1*, *MT3*, *VEGFA*
Cellular response to decreased oxygen levels (GO:0036294)	<0.001	*SLC8A3*, *BNIP3L*, *EGLN3*, *PTGIS*, *HILPDA*, *RORA*, *AK4*, *NDNF*, *NDRG1*, *MT3*, *VEGFA*
Positive regulation of cytokine production (GO:0001819)	<0.001	*SLC7A5*, *IL33*, *IL6*, *CADM1*, *LEP*, *SERPINE1*, *HILPDA*, *F2RL1*, *RORA*, *AGER*, *ELANE*
Positive regulation of p38MAPK cascade (GO:1900745)	0.0218	*LEP*, *VEGFA*
(b) Excerpted relevant KEGG terms		
Terms with upregulated genes following exposure to hypoxia	*p*-value	Genes
HIF-1 signaling pathway	<0.001	*LDHA*, *IL6*, *EGLN3*, *SERPINE1*, *SLC2A1*, *ENO2*, *VEGFA*
PI3K-Akt signaling pathway	0.0071	*IL6*, *EFNA3*, *DDIT4*, *COL4A6*, *PDGFB*, *THBS2*, *JAK3*, *PCK2*, *VEGFA*

GO, gene ontology; KEGG, Kyoto Encyclopedia of Genes and Genomes.

**Table 2 ijms-25-12599-t002:** Enrichment analysis of transcriptomic data (control vs. *DPP4* KD HLFs under normoxic conditions).

(a) Excerpted Relevant GO Terms		
Terms with Upregulated Genes Following *DPP4* Knockdown Under Normoxic Conditions	*p*-Value	Genes
Cellular response to hypoxia (GO:0071456)	<0.001	*SFRP1*, *HMOX1*, *AK4*, *MGARP*, *SCN2A*, *HIF1A*, *AQP1*
Cellular response to decreased oxygen levels (GO:0036294)	<0.001	*SFRP1*, *HMOX1*, *AK4*, *MGARP*, *SCN2A*, *HIF1A*, *AQP1*
Positive regulation of integrin-mediated signaling pathway (GO:2001046)	<0.001	*EMP2*, *NID1*, *LIMS2*
Extracellular matrix organization (GO:0030198)	0.0034	*POSTN*, *ELN*, *COL11A1*, *APLP1*, *SH3PXD2B*, *HAS2*, *NID1*, *ADAMTS7*
(b) Excerpted relevant KEGG terms		
Terms with upregulated genes following *DPP4* knockdown under normoxic conditions	*p*-value	Genes
cAMP signaling pathway	0.0034	*LIPE*, *OXTR*, *EDNRA*, *GABBR1*, *GIPR*, *HHIP*, *F2R*, *SSTR1*, *MAPK3*
cGMP-PKG signaling pathway	0.0299	*EDNRA*, *KCNJ8*, *ADRA1D*, *ADRA2C*, *SLC25A4*, *MAPK3*
ECM–receptor interaction	0.0347	*RELN*, *TNC*, *ITGA7*, *SDC1*

**Table 3 ijms-25-12599-t003:** Enrichment analysis of transcriptomic data (control and *DPP4* KD HLFs under hypoxic conditions).

(a) Excerpted Relevant GO Terms		
Terms with Upregulated Genes Following *DPP4* Knockdown Under Hypoxic Conditions	*p*-Value	Genes
Cellular response to hypoxia (GO:0071456)	<0.001	*SLC8A3*, *HMOX1*, *AK4*, *MGARP*, *HIF1A*, *MT3*, *AQP1*
Cellular response to decreased oxygen levels (GO:0036294)	<0.001	*SLC8A3*, *HMOX1*, *AK4*, *MGARP*, *HIF1A*, *MT3*, *AQP1*
Regulation of transforming growth factor beta production (GO:0071634)	0.0022	*TGFB2*, *ITGB8*, *HIF1A*
Regulation of transforming growth factor beta2 production (GO:0032909)	0.0041	*TGFB2*, *HIF1A*
Extracellular matrix organization (GO:0030198)	0.0101	*TGFB2*, *ELN*, *COL11A1*, *APLP1*, *HAS2*, *NID1*, *LOXL1*, *ADAMTS7*
(b) Excerpted relevant KEGG terms		
Terms with upregulated genes following *DPP4* knockdown under hypoxic conditions	*p*-value	Genes
MAPK signaling pathway	0.0115	*DUSP5*, *DUSP2*, *TGFB2*, *EFNA3*, *BDNF*, *RPS6KA1*, *KIT*, *TGFA*, *MET*, *MAPK3*
TGF-beta signaling pathway	0.0214	*TGFB2*, *ZFYVE9*, *TGFB3*, *GDF6*, *MAPK3*
cGMP-PKG signaling pathway	0.0233	*SLC8A3*, *KCNJ8*, *ADCY3*, *ADRB1*, *ITPR3*, *ADRA2C*, *MAPK3*

## Data Availability

The datasets presented in this study can be found online in the NCBI database (accession number: GSE281533).

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
