# Peer review of "Transcriptome Analysis of Fibroblasts in Hypoxia-Induced Vascular Remodeling: Functional Roles of CD26/DPP4"

_ijms, 2024, doi:10.3390/ijms252312599_

Round 1

Reviewer 1 Report

Comments and Suggestions for Authors

In the present work, the authors aimed to elucidate the role dipeptidyl peptidase-4 (DPP4) in the pathogenesis of hypoxic pulmonary hypertension using DPP4 knockout mice and in human lung fibroblast cells cultured under hypoxic conditions. The authors performed sequencing analysis of cultured cells RNA, retrieving several interesting results regarding transcriptome differences. The mansucript is well written, with good quality of English so it is easy to follow.

My questions are the following:

1) I have one major concern about statistical evaluation. First, all data should be tested whether they follow normal distribution or not. Student t-test could be used solely on samples that follow Gaussian distribution. Second, if data are normally distributed, two-way ANOVA should be performed as there are 2 variable factors in the studies (1: absence vs presence of functional DPP4 gene; 2: normoxia vs hypoxia). In case of Figure 4, these variable factors are WT/DPP4 KO and time (day 0 to day 14).

2) Please clarify why female and not male mice were used for the in vivo studies?

3) Please discuss why sample size is so different in BALF measurements? How did the authors calculate the needed sample size per group? There seem to be several outliers in these BALF measurements, I suggest to use the outlier identification feature of GraphPad Prism to improve data consistency. Besides, more samples in DPP4 KO groups would possibly lead to more statistical power.

4) I suggest to treat WT mice with the DPP4 inhibitor sitagliptin during the 4 weeks PH induction in order to evaluate its therapeutic potential in PH. This additional experiment with translational impact would be the only one supporting the last sentence of the abstract ("CD26/DPP4 may be a potential therapeutic target in patients with 27 PH associated with chronic hypoxia").

5) It is known that DPP4 plays an important role in T2DM pathogenesis. Is T2DM associated with higher risk for PH?

6) Reduced PPAR-gamma activity has been proven to contribute to PAH (Sci Transl Med. 2018 Apr 25;10(438):eaao0303. doi: 10.1126/scitranslmed.aao0303; Curr Opin Nephrol Hypertens. 2020 Mar;29(2):171-179. doi: 10.1097/MNH.0000000000000580). It would be interesting to evaluate PPARG gene and/or protein expression related to the loss of functional DPP4.

7) If the cGMP-PKG signaling is affected by the loss of DPP4 (Table 2), I would suggest testing the expression of PDEs, sGC and PKG in both mouse lungs and cultured human lung fibroblast samples as these proteins can be easily modified by pharmacons in later studies.

Reviewer 2 Report

Comments and Suggestions for Authors

The manuscript “Transcriptome Analysis of Fibroblasts in Hypoxia-Induced Vascular Remodeling: Functional Roles of CD26/DPP4” by Suzuki et al. addresses a relevant aspect of pulmonary vascular pathology, examining the impact of CD26/dipeptidyl peptidase-4 (DPP4) in hypoxia-induced pulmonary hypertension (PH) via transcriptomic analyses. The findings offer insights into fibroblast activation pathways, with implications for therapeutic targeting in chronic hypoxia-associated pulmonary diseases. The authors have meticulously prepared this article. Overall, the paper is well-prepared and written with concise language.

The authors are invited to add some additional mechanistic detail on how DPP4 specifically regulates the TGF-β signaling pathway under hypoxia would strengthen the manuscript.

The manuscript presents significant and promising findings on CD26/DPP4’s role in hypoxia-induced vascular remodeling. Hence, I recommend it for acceptance.

Round 2

Reviewer 1 Report

Comments and Suggestions for Authors

The authors addressed all my concerns during revision.